# Development of copper-catalyzed deaminative esterification using high-throughput experimentation

Yuning Shen [1], Babak Mahjour [1] & Tim Cernak [1]✉

Repurposing of amine and carboxylic acid building blocks provides an enormous opportunity to expand the accessible chemical space, because amine and acid feedstocks are typically low cost and available in high diversity. Herein, we report a copper-catalyzed deaminative esterification based on C–N activation of aryl amines via diazonium salt formation. The reaction was specifically designed to complement the popular amide coupling reaction. A chemoinformatic analysis of commercial building blocks demonstrates that by utilizing aryl amines, our method nearly doubles the available esterification chemical space compared to classic Fischer esterification with phenols. High-throughput experimentation in microliter reaction droplets was used to develop the reaction, along with classic scope studies, both of which demonstrated robust performance against hundreds of substrate pairs. Furthermore, we have demonstrated that this new esterification is suitable for late-stage diversification and for building-block repurposing to expand chemical space.

[1] Department of Medicinal Chemistry, University of Michigan, Ann Arbor, MI 48109, USA. ✉email: tcernak@med.umich.edu

Amines and carboxylic acids are abundant building blocks that are classically united via amide coupling[1–3]. While the amide coupling is a powerful reaction, there are hundreds of other hypothetical ways in which amines and carboxylic acids can be coupled, with each new transformation imprinting a unique physicochemical property fingerprint on the product[4]. An amine–acid esterification for instance would be a powerful complement to the amide coupling. Esters are one of the most prevalent functional groups among natural and industrial chemicals. Their synthesis has classically relied on Fischer's esterification method to unite alcohol and an acid[5], although complementary esterification reactions of acids with aryl halides[6–9], aryl boronates[10–12], aryl sulfonates[13], aryl iodonium salts[14,15] and silanes[16] have emerged. An amine–acid esterification would leverage the abundance of two popular building blocks (Fig. 1A). Aryl amines are frequently encountered in pharmaceutical research, so harnessing this functional group would also provide opportunities for late-stage diversification. Additionally, ester products are prevalent in agrochemicals, materials, fragrances, natural products, and pharmaceuticals (Fig. 1B) such as camostat (1), gabexate (2), and candoxatril (3). As pharmaceuticals, esters are commonly used as prodrugs, as in 2, or as short-acting agents like local anesthetics. While many, but not all, esters are readily cleaved by plasma esterases[17], several medicines, such as taxol or 1, highlight the direct role esters can play in inducing diverse bioactivities.

Recently C–N bond activation has emerged as an important synthetic strategy to exploit the commercial and natural prevalence of the amine functional group[18–21], leading us to consider the transformation of a C–N bond into a C–O bond. A handful of C–N to C–O transformations have been reported[22–30], but this transformation class remains largely unexplored. We were excited to extend von Pechmann's classic methyl esterification with diazomethane[31] to a mild, selective, and robust aryl ester synthesis employing aryl diazonium salts. While diazonium salts may present an explosion hazard[32–34], these reactive species can be handled with proper precautions or in flow[35,36]. Our approach is to target reactions through chemoinformatic analysis and prosecute the discovery and development using high-throughput experimentation (HTE)[4,37]. On the small reaction scale employed in miniaturized HTE[38,39], the hazards of handling diazonium salts are minimized. For esterification, we performed exploratory screens of diverse reaction spaces in 24 wells, followed by systematic reaction optimization in 96 wells, and finally reaction performance profiling in 1536 wells (Fig. 1C). Thus, our method provides drug hunters a means to repurpose their chemical building-block libraries, making amide products in the traditional approach or ester analogs using our approach.

Anilines are a cheap, abundant feedstock and are commercially available in high diversity, making them a valuable starting material for ester synthesis. By repurposing aniline building blocks as esters, instead of the classic amide, a subtle change in physicochemical properties emerges (Fig. 2A). Esters have fewer hydrogen bond donors (HBD) than amides (Fig. 2A), potentially increasing the permeability of the products across biological membranes[40]. We compared a virtual library of esters and amides, derived from the drug metoclopramide (Fig. 2D and Supplementary Information), for their predicted blood–brain permeability using the central nervous system probabilistic multiparameter optimization (CNS-pMPO) score[40,41]. The average and maximum predicted blood–brain barrier permeability was much increased for ester products versus amide products generated from the same building blocks. We also performed a survey of commercially available building blocks from the MilliporeSigma catalog (Fig. 2B), which revealed that there are 4142 unique phenols and 2996 unique anilines, with only 633 matched molecular pairs between the two

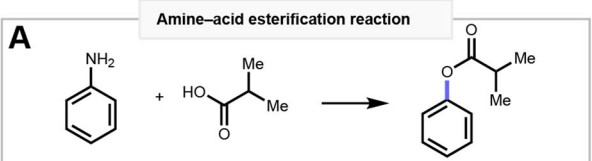

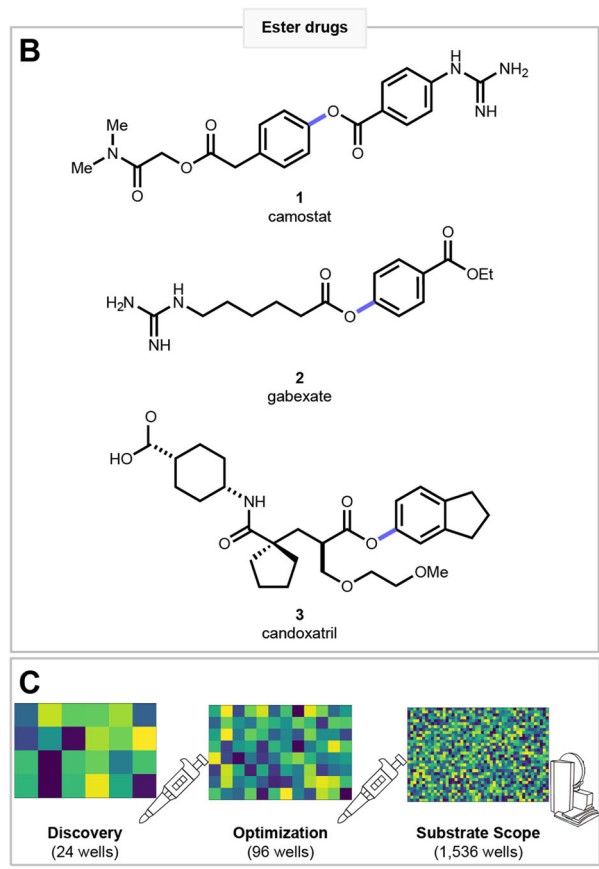

**Fig. 1 HTE enabled amine–acid esterification and its application in drug molecules. A** The amine–acid esterification. **B** Esters are an important medicinal functionality as in drugs **1**–**3**. **C** Reaction discovery, optimization, and scope profiling is achieved using HTE.

sets of building blocks. This 42% expansion of accessible chemical space from the anilines, compared to the phenols, can be readily seen in a T-distributed Stochastic Neighbor Embedding (tSNE) analysis of computationally enumerated esters produced from aspirin using either the 4142 phenols via Fischer esterification or the 2996 anilines using our amine–acid esterification (Fig. 2C). There is minimal overlap of chemical space, demonstrating that an amine–acid esterification can provide broad access to new and complementary structures. Collectively, these analyses quantify the value that an amine–acid esterification would provide as an addition to the synthetic toolbox.

## Results and discussion

We reasoned that transition metals capable of both activating the C–N bond of a diazonium salt and forming a C–O bond would be capable of achieving the desired esterification (Fig. 3A). Recent studies employ diazonium salts as substrates in palladium[42–47], copper[48–50], and gold[51] catalyzed Meerwein-type, Suzuki-type, and Sonogashira-type coupling reactions. There has also been a resurgence of Sandmeyer-type reactions forging C–heteroatom bonds from diazonium salts to leverage the low cost of aryl amines[33,52,53]. Since palladium, nickel, and copper have been

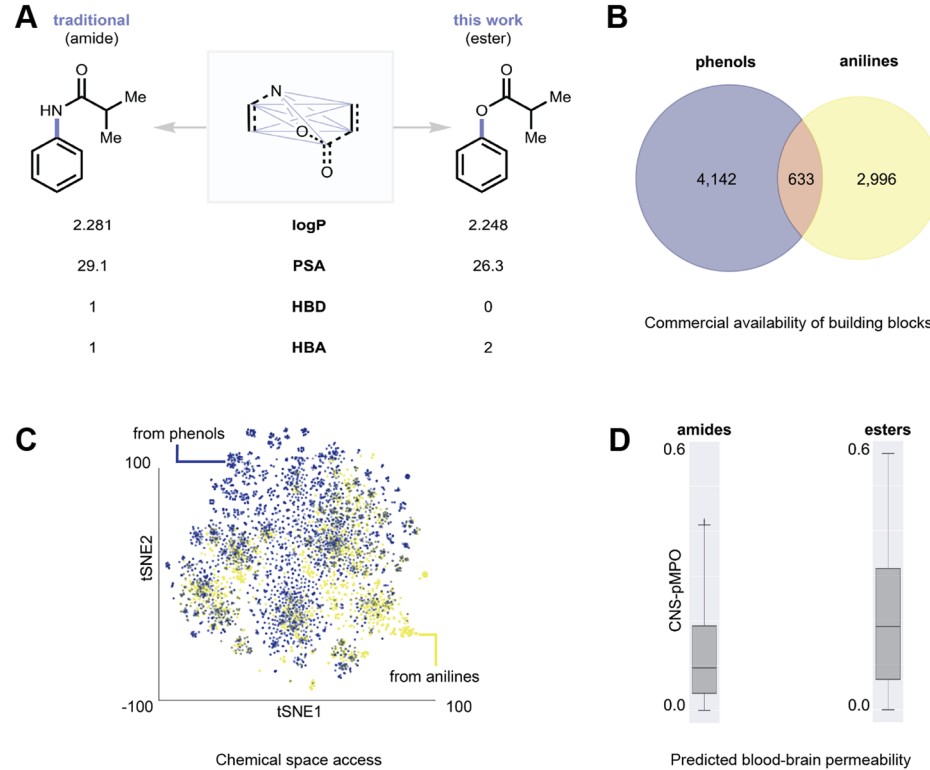

**Fig. 2 Physicochemical properties of amides and esters and complementary chemical space of anilines to phenols. A** An amine and carboxylic acid can be coupled in a variety of transformations beyond the traditional amide coupling, including the esterification reaction developed here, which gives a unique property footprint. **B** Venn diagram showing the complementarity of phenol to aniline building blocks available in the MilliporeSigma catalog. **C** tSNE analysis showing the complementarity of ester products generated by coupling aspirin to phenol (blue dots) versus aniline (yellow dots) building blocks available in the MilliporeSigma catalog. **D** Ester products have a higher predicted blood–brain permeability (CNS-pMPO score) than classic amide products produced from the same set of anilines when coupled to the drug metaclopramide.

used for both diazonium activation and C–O coupling, we interrogated these metals' ability to forge esters from diazonium salts using HTE in 24 wells (Fig. 3B). As a preliminary result, we identified that the combination of one equivalent each of copper iodide, silver nitrate, and pyridine promotes the esterification of **4** with **5** to give **6**. This data, which was included in our first amine–acid coupling report[4], demonstrated the feasibility of the desired reaction but required the use of two metals in stoichiometric amounts. Subsequently, we surveyed a variety of copper salts and ligands in 96-well arrays with **4** and **5** producing **6** under catalytic conditions. A first survey identified (CuOTf)$_2$•C$_6$H$_6$ with no ligand as the best condition (Fig. 3C) when 2,4,6-collidine was used as a base. A deeper survey of ligands and bases confirmed that diverse ligands such as phosphines, bipyridines, diamines, and oxalamides (see Supplementary Information) were detrimental to reaction progression. Likewise, 2,4,6-collidine remained the optimal base. Spectroscopic studies (Fig. 3E) supported the hypothesis that collidine was acting as a base, rather than a ligand. This was evidenced by a bathochromic shift only observed when copper, acid, and collidine were mixed, suggesting the carboxylate thus formed coordinates with copper. Select results from miniaturized reactions were repeated, alongside additional optimization conditions, on a 0.300 mmol reaction scale and are presented in Fig. 3F. The nature of the copper salt used had a modest impact on reaction performance (entries 1–6) and we moved forward with Cu(MeCN)$_4$BF$_4$ as a preferred promoter because it displayed optimal performance and is common and affordable (entry 3). Reduction of the copper loading below 30 mol% was viable although yields were lower. Since no expensive ligand is required

for the reaction, and the copper salt used is itself relatively inexpensive, we elected to move forward with 30 mol% Cu(MeCN)$_4$BF$_4$ with 2,4,6-collidine in acetonitrile as our preferred conditions. No product was formed in the absence of base (entry 9), Alternate bases gave sub-optimal performance (entries 6–9) and copper salt was required for the reaction to occur (entry 10). Acetonitrile was the optimal solvent (entry 3 versus 11–13), although **6** was observed with other nitrile solvents, such as benzonitrile (entry 11). We sought to explore the feasibility of this esterification for late-stage diversification using heterocyclic and pharmaceutically relevant aryl amines. We chose to explore complex molecule diversification using miniaturized ultraHTE in 1536 well plates[54]. In this format, reactions are executed by nanoliter robotic dosing in an inert atmosphere glovebox, in a plastic microtiter plate[55–59], with reaction analysis determined by assay yield, or conversion to product relative to an internal standard[59] using UPLC-MS. Nanomole-scale reactions in 1536-well plates have not yet been reported outside of an industrial setting. To validate this technique in an academic setting, we performed quadruplicate entries for 384 diazonium–acid substrate pairs (1536 reactions in total) thus interrogating the reproducibility of the method. Diazonium salts (**7–10**, Fig. 3G) were prepared from 8-aminoquinoline, sulfadoxine, sulfamethoxazole, and metoclopramide respectively, then dosed into 1536-well plates on an SPT Labtech mosquito® under a nitrogen atmosphere. To facilitate reaction miniaturization, we used benzonitrile in place of acetonitrile as a high boiling solvent. Since we anticipated the formation of the product would be lowered in this solvent (Fig. 3F, entry 11), we raised the loading of Cu(MeCN)$_4$BF$_4$ to 100 mol%. Among the 384 pairs of substrates

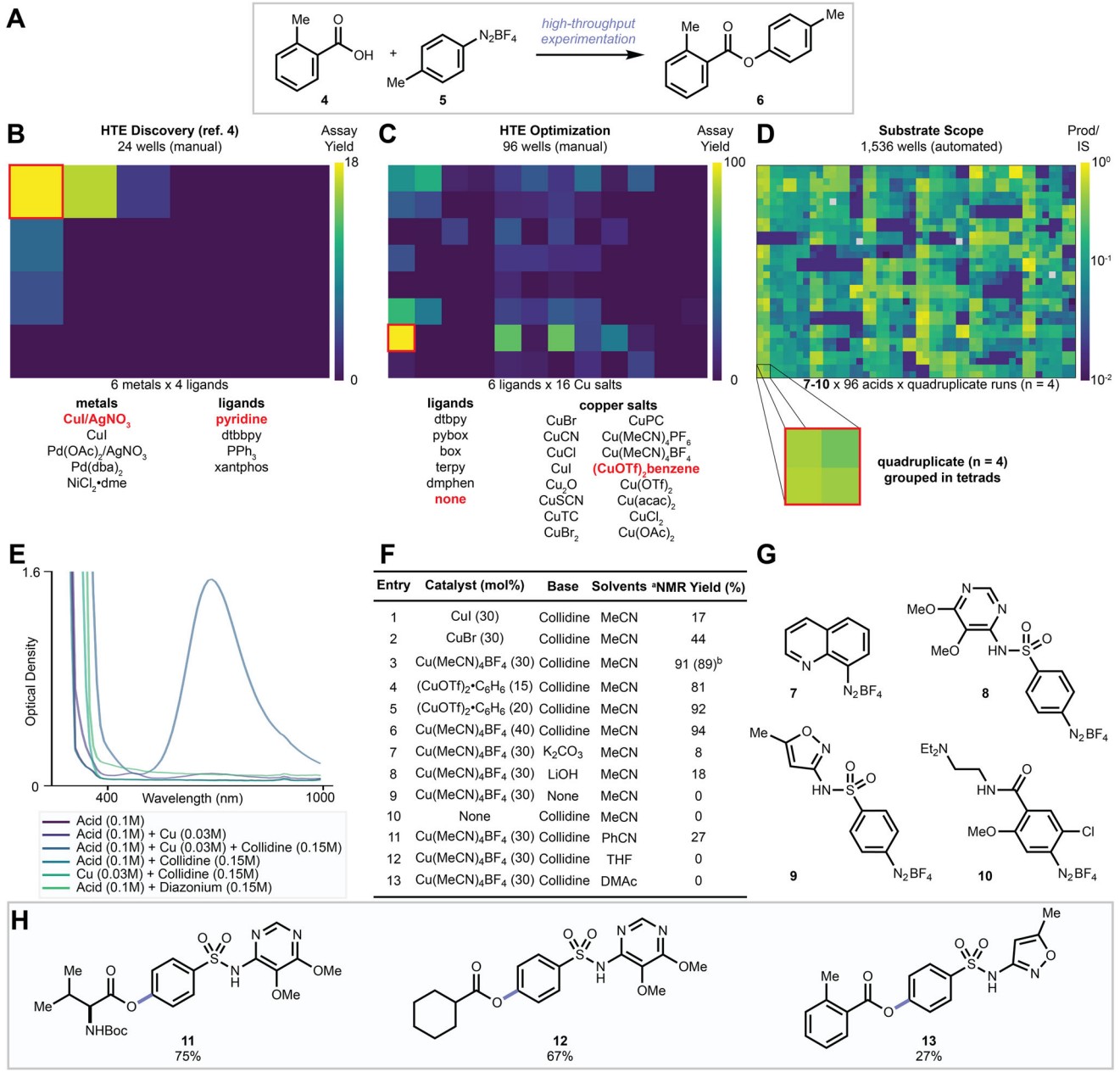

**Fig. 3 Reaction discovery and profiling with HTE. A** General deaminative esterification reaction. **B** Heatmap showing the discovery of the esterification reaction in 24 glass microvials. Dtbpy = 4,4′-di-*tert*-butyl-2,2′-dipyridyl, xantphos = 4,5-bis(diphenylphosphino)-9,9-dimethylxanthene. **C** Heatmap showing optimization of esterification in 96 glass microvials (see Supplementary Information for details). Pybox = 2,6-bis[(4 S)-( − )-isopropyl-2-oxazolin-2-yl]pyridine, box = 2,2-Bis((4 S)-(-)-4-isopropyloxazoline)propane, terpy = 2,2′:6′,2′′-terpyridine, dmphen = 4,7-dimethoxy-1,10-phenanthroline, CuTC = copper(I) thiophene-2-carboxylate, CuPC = copper(II) phthalocyanine. **D** Heatmap showing the reproducibility of quadruplicate data for a library of 96 acids coupled to **7**–**10** in plastic 1536-well plates. The overall average standard deviation across the quadruplicate data was 5.5%. The wells with the gray color indicated that the dosage of the internal standard was missing. **E** UV-Vis absorbance data show an interaction between Cu(MeCN)₄BF₄, **4** and 2,4,6-collidine. **F** Reaction optimization. a ¹H-NMR yield with 1,3,5-trimethoxybenzene as internal standard, b Isolated yield. **G** Diazonium salts used in the 1536 screen shown in **D**. **H** Select reactions from those shown in **D** were repeated on a 0.300 mmol scale using Cu(MeCN)₄BF₄ (100 mol%) to produce **11**, **12**, and **13** in isolated yield shown.

surveyed, the desired ester products were reproducibly observed by UPLC-MS analysis in 322 instances (Fig. 3D) with an average standard deviation of 5.5% across the quadruplicate data (see Supplementary Information). Informed by this ultra-HTE study, late-stage diversification reactions were performed on a 0.300 mmol scale, giving **11**–**13** in 27–75% isolated yield (Fig. 3H).

The substrate scope for this reaction is very broad. We explored the generality of coupling diazonium salts to a series of carboxylic

acids (Fig. 4). Aliphatic (**17**, **24**, **36**), benzylic (**23**, **32**, **37**), and aromatic (**6**, **27**, **35**) carboxylic acids performed well giving desired products in 48–89% isolated yield. Amino acids were viable substrates, with both N-Boc (**7**) and N-tosyl proline (**31**) generating the desired ester product (**8**, **24**) in good yield. A variety of functionalities were tolerated such as N-Boc groups (**5**, **20**, **3**, **4**), *N*-toluenesulfonamides (**18**, **31**), nitro- (**27**), thia-zolo- (**30**), fluoro- (**23**), and styrenyl motifs (**25**). Diazonium salts derived from both electron-rich (**6**, **16**, **19**, **38**) and electron-

**Fig. 4 Substrate scope.** Reactions were run with carboxylic acids (0.300 mmol), diazonium salts (1.5 equiv.), Cu(MeCN)$_4$BF$_4$ (30 mol%), and 2,4,6-collidine (1.5 equiv.) in anhydrous acetonitrile (0.1 M) at room temperature, generally over 16 h. Reported yields are isolated yields of purified products.

deficient (**21**, **26**, **28**) aryl amines were successful giving desired products in 64–89% yield. Sterically hindered diazonium salts were also viable (**34**). Alkyl diazonium salts were not attempted in this chemistry owing to their perceived instability.

Having demonstrated a robust substrate scope for this C–N to C–O transformation, we considered a variety of applications. For

instance, the ester **24**, which was produced in 76% yield, is a potential intermediate in the synthesis of the marketed protease inhibitor gabexate (**2**). Recognizing that in situ activations of amine substrates would be operationally convenient[60], we explored an activation strategy (See Supplementary Information) wherein $^i$AmONO and BF$_3$•Et$_2$O were added to the free aniline, p-toluidine,

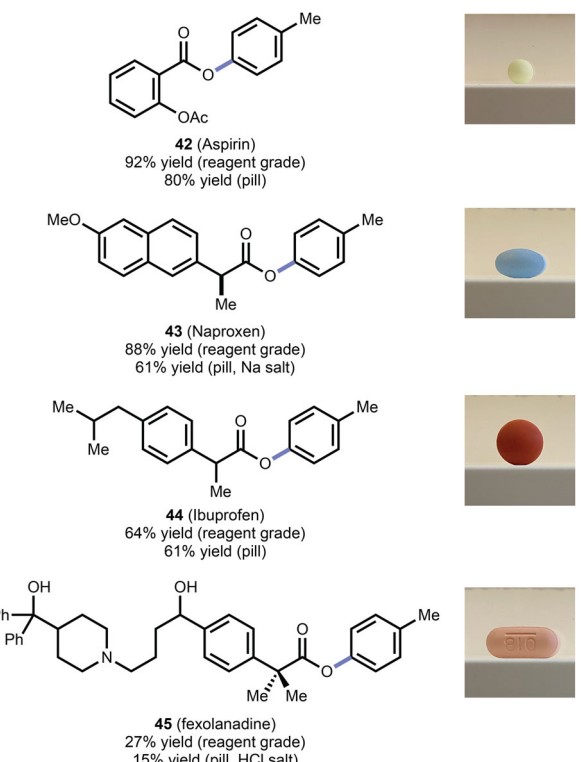

**42 (Aspirin)**
92% yield (reagent grade)
80% yield (pill)

**43 (Naproxen)**
88% yield (reagent grade)
61% yield (pill, Na salt)

**44 (Ibuprofen)**
64% yield (reagent grade)
61% yield (pill)

**45 (fexolanadine)**
27% yield (reagent grade)
15% yield (pill, HCl salt)

**Fig. 5 The amine–acid esterification applied directly to over-the-counter pills.** Ester products **42**–**45** are derived from both reagents and OTC pills (See Supplementary Information for details).

prior to the addition of the other reagents. By this protocol, the desired ester product was isolated in 72% yield, compared to 89% for the preparation of **6** from the isolated diazonium salt **5**.

Finally, to stress-test reaction performance, we evaluated the robustness of this reaction towards pharmaceutical diversification on actual pharmaceuticals obtained from crushed over-the-counter pills (Fig. 5). The reaction was successful regardless of whether the carboxylic acid was reagent grade or obtained from the crushed pill, with no additional treatment to remove fillers or excipients, giving esters derived from aspirin (**42**), naproxen (**43**), ibuprofen (**44**), and fexofenadine (**45**) in comparable yield and demonstrating that even substrates contaminated with pill excipients are viable. Remarkably, the basic amine and free alcohols of 45 were tolerated, albeit in somewhat reduced yield. The ability to use free alcohols highlights the complementarity of our method with classic alcohol–acid esterification protocols.

## Conclusion

In conclusion, we have developed copper-catalyzed esterification of diazonium salts with carboxylic acids and demonstrated its use in library synthesis with ultraHTE. This C–N to C–O conversion is a complement to the traditional amide coupling, which generates products with similar shapes and electronics but with one fewer hydrogen bond donor (HBD). The new coupling reaction reported here thus adds to a growing menu of amine–acid transformations that can be selected to modulate physicochemical properties.

## Methods

All reactions were conducted in the oven- or flame-dried glassware under an atmosphere of nitrogen unless stated otherwise. Reactions were set up in an MBraun LABmaster Pro Glove Box (H$_2$O level <0.1 ppm, O$_2$ level <0.1 ppm), or using the standard Schlenk technique with a glass vacuum manifold connected to an inlet of dry nitrogen gas. Solvents (acetonitrile, tetrahydrofuran,

dichloromethane) were purified using a MBraun SPS solvent purification system, by purging with nitrogen, and then passing the solvent through a column of activated alumina. Flash chromatography was performed on silica gel (230–400 Mesh, Grade 60) under a positive pressure of nitrogen. Thin Layer Chromatography was performed on 25 μm TLC Silica gel 60 F$_{254}$ glass plates purchased from Fisher Scientific (part number: S07876). Visualization was performed using ultraviolet light (254 nm), potassium permanganate (KMnO$_4$) stain. See Supplementary Information for additional details. Data used to produce the heatmap is also available in the Supplementary Information. Diazonium salts may exhibit explosion hazards, especially when used in anhydrous solid form, so they should be handled with caution. Refs. [32–34] provide additional details for the safe handling of diazonium salts. The diazonium salts in this work were stored in the freezer in a secondary container and used within one month after preparation.

## Data availability

Raw data and experimental procedures are available in the Supplementary Information. The general methods for the preparation of ester products **6**–**45** and their characterizations are available in the Supplementary Information (Section Supplementary Methods), ¹H-NMR, ¹³C NMR spectra for compounds **6**–**45** and ¹⁹F NMR spectra for fluorine-containing compounds are available in the Supplementary Information (Section Supplementary Note 1: Spectra). Additionally, preparation for diazonium salts and HTE operation referred to literature references in Supplementary References.

## Code availability

All code for the chemoinformatic portion of this study can be found at https://github.com/cernaklab/acid-amine-esterification.

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

## Acknowledgements
The authors wish to thank the University of Michigan College of Pharmacy for start-up funds. Dr. Amie Frank is thanked for their assistance in the preparation of this manuscript.

## Author contributions
Y.S. designed and executed chemistry experiments, processed, and analyzed the high-throughput experimentation data. B.M. performed chemoinformatic studies. All authors reviewed the data and wrote the manuscript. T.C. supervised the study.

## Competing interests
The Cernak Lab has received research funding or in-kind donations from MilliporeSigma, Relay Therapeutics, Janssen Therapeutics, SPT Labtech, and Merck & Co., Inc. T.C. holds equity in Scorpion Therapeutics, and is a co-Founder and equity holder of Entos, Inc.
