## [Peer Review File · Communications Chemistry]

Reviewers' comments:

Reviewer #1 (Remarks to the Author):

Cernak and co-workers report a copper-catalyzed esterification of carboxylic acids with aryl diazonium salt with $\text{Cu}(\text{MeCN})_4\text{BF}_4$ (30 mol%) as a catalyst, and 2,4,6-collidine (1.5 equiv.) as a base under room temperature. Compared with other esterification methods that use aryl halides, aryl boronates, aryl sulfonates, aryl iodonium salts and silanes as aryl source, the current method uses aryl diazonium salt as aryl reagents, allowing a broader range of aryl source, because arylamines are readily available and inexpensive materials. Due to great abundance of both carboxylic acids and amines, this method provides a possibility for the construction of a diverse range of esters. In addition, the authors used high throughput experimentation (HTE) technique for reaction optimization and scope investigation, achieving high efficiency. The limitations of this work lie in high loadings of Cu catalysts, anhydrous conditions, limited scope of aryl diazonium salts, and as well as a two-step protocol from amines. However, on the basis of the novelty and good applicability, this manuscript is still recommended for publication after addressing the following questions.

1. Owing to only one example on one-pot esterification using in-situ diazotization, the title "deaminative esterification" is not correct.
2. Page 7, Fig. 4, the reaction equation is not correct. Compound 5 should be drawn as ArN_2BF_4 .
3. Are alkyl diazonium salts compatible with this reaction, if not, the authors should add a note in the text.

Reviewer #2 (Remarks to the Author):

Shen, Mahjour, and Cernak present high-quality work focused on the deaminative esterification of aryl amines and carboxylic acids. As the authors note, this reaction was published by the Cernak lab in a prior work as part of the, but this manuscript focuses on further optimization and substrate scope, as well as how the transformation might be used in applied settings (drug discovery or other industries that need esters). While esters are not as stable as amides (amides are often used in drug discovery for this exact reason), the authors present convincing arguments why this work is important. The manuscript shows how the team used high throughput experimentation (HTE) in various forms, both manual and automated, to optimize the reaction and broadly explore the substrate scope. One of the most important aspects of this work, in my view, is the demonstration of translation of nano-scale liquid handling and reaction execution in 1536 well plates on small scale, usually performed in industry, to an academic laboratory. This is a large forward step for those who want to explore chemical and reaction space. I am curious to see if and when the total synthesis community takes advantage of this technology. Looking forward to what else comes of this. The SI looks to be order.

Here are a few suggestions for the content/data of the paper:

- 1) The authors describe use of 100 mol% Cu for nano-scale experiments, because it was anticipated that product formation would be lower in benzonitrile solvent. Please perform a control to show this (or not), because this control experiment links the nano-scale scope experiments to the reactions performed on scale with isolated or otherwise measured yields.
- 2) In figure 3, there a few heat maps that need more detail. The way they are presented is clean, but the reader is forced to browse the SI to get key information. I'll leave it up to the authors to decide how to this, but as is, these are heat maps with minimal data on the axis that don't mean much to the reader, especially to those not versed in data presented in this format.
- 3) The 1536 heat map in figure 3 needs some explanation too. Readers who are not nano-chemistry experts won't understand that the quadruplicate runs are all grouped together, unless this is pointed

out. The data is structured in a way that an expert can clearly see that there are squares of the same color, each corner of the square is one smaller square that represents a single run of the set, that are grouped together, but non-experts won't know this. Maybe make a pop-out image that shows how the data/plate layout is structured. This discussion should be included in the text, or the Figure 3 caption, or at the very least, a citation added directing readers to where this information is presented elsewhere.

4) Were any of the examples in the nano-experiment presented in figure 4? If so, they should be highlighted to demonstrate translation and help users calibrate results presented that were generated on nanoscale.

5) What is the value of adding caffeine IS after the nano-chemical experiment is complete as the calibration of the products is not practical on this number of products? If it were added with the diazonium stocks and a sample was measured before chemistry executed, quantitative conversion to product could be measured per product couple, however as I read the SI this was not done. Is the outcome different if you don't use Prod/IS and just area percent? Shouldn't these values be proportional? Is the standard relevant because benzonitrile is so UV-active making LCAP an unusable metric? As caffeine is used, how do you know that the product peaks and the caffeine standard don't overlap? It would be nice if the authors can confirm that most of the products retain longer or shorter on the LC column than caffeine and add this information to the SI. Or maybe there is a high-level data visualization that can show this. If retention time was captured, maybe a scatter plot with retention time of standard and product on the y-axis vs. each reaction. The data would be structured such that the 4 points (per each reaction) are separated in the y direction from the standard, if there is no overlap. If they are tight, then there likely is.

At the very least, maybe include a few representative HPLC traces in the SI.

6) I don't see the value of the experiments described in figure 5 without describing what the other materials in the pharmaceutical pills are. This may be common knowledge to some, but likely not all readers. Please comment in the text to bolster this part. The more impressive facet of this section is example 45. On this note, it would be great if the authors included results that highlight negative data in their scope table. I realize this is in the nano-heat map and associated data. Maybe a few examples that gave no conversion in the heatmap added to figure 4.

7) The authors describe an in-situ activation of p-tol-aniline using isoamyl nitrite and BF₃ etherate. This is a useful addition for practitioners, especially people in applied settings who are going to prepare and handle diazoniums. On this note, the authors should comment explicitly on the safety and practical limitations that exist when using diazoniums, a foot note is good enough. The tetra-fluoro borate diazoniums tend to be stable but addressing this directly may help uptake and help those who don't know about the hazards of different types to stay safe.

8) Here is a suggested experiment that would elevate this manuscript: As the in-situ generation of diazoniums followed by esterification works, it would be great if the authors could demonstrate this in parallel (if it works) with an array of aryl amines. Even better would be to cross that with a few acids. This is fine to do with manual pipetting (maybe 12 amines and 8 acids). This type of experiment would be what people in a pharmaceutical medicinal setting often perform, varying two coupling partners at once in a 2-dimensional library synthesis.

Suggested edits to the text:

in introduction

1st paragraph "...there are hundreds of other hypothetical ways amines and carboxylic acids can be coupled..."

1st paragraph "Aryl amines are frequently encountered in pharmaceutical research, so harnessing this functional group would also..."

Last sentence in introduction "Collectively, these analyses highlight quantitatively the value that an amine-acid esterification would provide as an addition to the synthetic toolbox.

In results and discussion

2nd paragraph "The substrate scope for this reaction is broad"

Reviewer #3 (Remarks to the Author):

Shen, Mahjour, and Cernak disclose a method to O-arylate carboxylic acids via the use of diazonium salts. A copper catalyst is used, giving the method some similarities to Chan Lam oxidative coupling reactions or copper-mediated Sandmeyer reactions. The developed method is relatively straightforward, occurring in quite high yield with a good scope under mild reaction conditions. A high throughput examination of scope is carried out, with four different diazonium salts being coupled with 96 acids (384 different products), all run in quadruplicate. Select hits are scaled up to provide useful quantities of product and validate the high throughput evaluation method.

The manuscript is well written and fairly easy to read despite the large amount of data disclosed from the HTE data. The science also appears rigorously executed and high quality. There are certainly limitations with the use of diazoniums as coupling partners, but the broad scope investigated suggests that it may on occasion be worth the effort. The incorporation of HTE into the method development and scope evaluation is also seldom done and of general interest. As such, publication in Communications Chemistry is recommended as-is.

May 5th, 2022

Dear Editor and Reviewers:

We thank you kindly for the very thoughtful and helpful feedback on our manuscript "Development of Deaminative Esterification Using High-Throughput Experimentation" for *Communications Chemistry*. Our response to the constructive suggestions is shown below in blue.

Reviewers' comments:

Reviewer #1 (Remarks to the Author):

Cernak and co-workers report a copper-catalyzed esterification of carboxylic acids with aryl diazonium salt with Cu(MeCN)₄BF₄ (30 mol%) as a catalyst, and 2,4,6-collidine (1.5 equiv.) as a base under room temperature. Compared with other esterification methods that use aryl halides, aryl boronates, aryl sulfonates, aryl iodonium salts and silanes as aryl source, the current method uses aryl diazonium salt as aryl reagents, allowing a broader range of aryl source, because arylamines are readily available and inexpensive materials. Due to great abundance of both carboxylic acids and amines, this method provides a possibility for the construction of a diverse range of esters. In addition, the authors used high throughput experimentation (HTE) technique for reaction optimization and scope investigation, achieving high efficiency. The limitations of this work lie in high loadings of Cu catalysts, anhydrous conditions, limited scope of aryl diazonium salts, and as well as a two-step protocol from amines. However, on the basis of the novelty and good applicability, this manuscript is still recommended for publication after addressing the following questions.

We thank the Reviewer for this supportive feedback, and address their concerns below:

1. Owing to only one example on one-pot esterification using in-situ diazotization, the title "deaminative esterification" is not correct.

We understand the Reviewer's sentiment here, but choose to leave the title as is in line with other contemporary reports on deamination using Katritzky salts and related C–N activating protocols.

2. Page 7, Fig. 4, the reaction equation is not correct. Compound 5 should be drawn as ArN_2BF_4 .

We thank the reviewer for pointing this out, Compound 5 was redrawn.

3. Are alkyl diazonium salts compatible with this reaction, if not, the authors should add a note in the text.

We thank the Reviewer for this clarifying suggestion. The following text was added: "Alkyl diazonium salts were not attempted in this chemistry owing to their perceived instability"

Reviewer #2 (Remarks to the Author):

Shen, Mahjour, and Cernak present high-quality work focused on the deaminative esterification of aryl amines and carboxylic acids. As the authors note, this reaction was published by the Cernak lab in a prior work as part of the, but this manuscript focuses on further optimization and substrate scope, as well as how the transformation might be used in applied settings (drug discovery or other industries that need esters). While esters are not as stable as amides (amides are often used in drug discovery for this exact reason), the authors present convincing arguments why this work is important. The manuscript shows how the team used high throughput experimentation (HTE) in various forms, both manual and automated, to optimize the reaction and broadly explore the substrate scope. One of the most important aspects of this work, in my view, is the demonstration of translation of nano-scale liquid handling and reaction execution in 1536 well plates on small scale, usually performed in industry, to an academic laboratory. This is a large forward step for those who want to explore chemical and reaction space. I am curious to see if and when the total synthesis community takes advantage of this technology. Looking forward to what else comes of this. The SI looks to be order.

We thank the Reviewer for their kind words on our work – it was indeed a major challenge for us to build the 1536 capabilities in our new lab. We hope this will be the first report of many using the technology for reaction development.

Here are a few suggestions for the content/data of the paper:

1) The authors describe use of 100 mol% Cu for nano-scale experiments, because it was anticipated that product formation would be lower in benzonitrile solvent. Please perform a control to show this (or not), because this control experiment links the nano-scale scope experiments to the reactions performed on scale with isolated or otherwise measured yields.

We thank the reviewer for pointing out the changes in copper catalysts loading. In the optimization table (Figure 3 F, entry 11) we have tried the reaction using 30 mol% Cu in benzonitrile, which gave 27% NMR yield.

2) In figure 3, there a few heat maps that need more detail. The way they are presented is clean, but the reader is forced to browse the SI to get key information. I'll leave it up to the authors to decide how to this, but as is, these are heat maps with minimal data on the axis that don't mean much to the reader, especially to those not versed in data presented in this format.

We made significant edits to the figure to further clarify the makeup of the screens and the winning hits, ultimately we are presenting >1600 experiments in a very small amount of page space. While there are limits to how many details can be included in a small figure, we hope the updated version will be more accessible to our readers who are new to interpreting HTE data.

3) The 1536 heat map in figure 3 needs some explanation too. Readers who are not nano-chemistry experts won't understand that the quadruplicate runs are all grouped together, unless this is pointed out. The data is structured in a way that an expert can clearly see that there are squares of the same color, each corner of the square is one smaller square that represents a single run of the set, that are grouped together, but non-experts won't know this. Maybe make a pop-out image that shows how the data/plate layout is structured. This discussion should be included in the text, or the Figure 3 caption, or at the very least, a citation added directing readers to where this information is presented elsewhere.

We thank the Reviewer for this excellent suggestion. A pop-out box was added showing that the quadruplicate reactions are grouped together.

4) Were any of the examples in the nano-experiment presented in figure 4? If so, they should be highlighted to demonstrate translation and help users calibrate results presented that were generated on nanoscale.

Compound 33 (isolated in 66% yield (Figure 4) was screened in the 1,536-uHTE experiment, with an average LC ratio of 36.5%.

5) What is the value of adding caffeine IS after the nano-chemical experiment is complete as the calibration of the products is not practical on this number of products? If it were added with the diazonium stocks and a sample was measured before chemistry executed, quantitative conversion to product could be measured per product couple, however as I read the SI this was not done. Is the outcome different if you don't use Prod/IS and just area percent? Shouldn't these values be proportional? Is the standard relevant because benzonitrile is so UV-active making LCAP an unusable metric? As caffeine is used, how do you know that the product peaks and the caffeine standard don't overlap? It would be nice if the authors can confirm that most of the products retain longer or shorter on the LC column than caffeine and add this information to the SI. Or maybe there is a high-level data visualization that can show this. If retention time was captured, maybe a scatter plot with retention time of standard and product on the y-axis vs. each reaction. The data would be structured such that the 4 points (per each reaction) are separated in the y direction from the standard, if there is no overlap. If they are tight, then there likely is.

At the very least, maybe include a few representative HPLC traces in the SI.

We thank the reviewer for this question. Caffeine (1.0 equiv) is the internal standard added in the quenching DMSO solution after the HTE in 1,536 well plate was finished. The addition of the internal standard was operated while dividing the reaction crudes into four 384- analytical well plates, which is typically done to confirm there are no dosing errors. To be more consistent with the operational protocol, we used product over internal standard ratio in visualization. In addition UV absorption vary in a broad range for different products, the LC ratio of product over internal standard will be easier to accommodate such big differences. Caffeine is used as the internal standard because in our LC method, caffeine has a retention time at 0.44 min in the early retention time window, while most of our products retain later than caffeine. Some representative UPLC traces were included in SI as Figure S7 for clarification. These analysis tactics are standard in the HTE community (cf. Ref 59).

6) I don't see the value of the experiments described in figure 5 without describing what the other materials in the pharmaceutical pills are. This may be common knowledge to some, but likely not all readers. Please comment in the text to bolster this part. The more impressive facet of this section is example 45. On this note, it would be great if the authors included results that highlight negative data in their scope table. I realize this is in the nano-heat map and associated data. Maybe a few examples that gave no conversion in the heatmap added to figure 4.

We thank the Reviewer for their feedback, and indeed appreciate that running reactions on pills is atypical. In large part that is the point. Many of our peers choose to run reactions in tea or beer to showcase the robustness of their methods, and we find such examples instructive. As we are in the College of Pharmacy and many in our community work with formulated pills, this experiment was a fitting example for our lab to run. We include the pill sources and lot numbers in the SI. We agree that example 45 is a good result and thank the Reviewer for pointing it out. In Figure 4, we don't have any paper real estate to include more examples but as noted we disclose many many more negative datapoints than most methods papers through our HTE data.

7) The authors describe an in-situ activation of p-tol-aniline using isoamyl nitrite and BF₃ etherate. This is a useful addition for practitioners, especially people in applied settings who are going to prepare and handle diazoniums. On this note, the authors should comment explicitly on the safety and practical limitations that exist when using diazoniums, a foot note is good enough. The tetra-fluoro borate diazoniums tend to be stable but addressing this directly may help uptake and help those who don't know about the hazards of different types to stay safe.

We thank the reviewer for pointing out the safety concerns of handling diazonium salts. We added notes in the method section to call readers' attention to potential hazards of handling diazonium salts, and many details of mitigating these hazards on larger reaction scale are described in references 32-34. We also added more practical details for the esterification reaction in the SI.

8) Here is a suggested experiment that would elevate this manuscript: As the in-situ generation of diazoniums followed by esterification works, it would be great if the authors could demonstrate this in parallel (if it works) with an array of aryl amines. Even better would be to cross that with a few acids. This is fine to do with manual pipetting (maybe 12 amines and 8 acids). This type of experiment would be what people in a pharmaceutical medicinal setting often perform, varying two coupling partners at once in a 2-dimensional library synthesis.

We have tried in-situ diazotization HTE screening 24 anilines to couple with one carboxylic acid. Unfortunately, giving the nature of the diazonium salts even in-situ formed, it's difficult to operate the in-situ protocol in HTE format with low temperature in inert atmosphere with exclusion of moisture. While the in situ protocol works in a singleton format, we have not had a satisfactory result in a HTE screening format although we are actively engaged in achieving this result in follow up studies.

Suggested edits to the text:

in introduction

1st paragraph "...there are hundreds of other hypothetical ways amines and carboxylic acids can be coupled..."

1st paragraph "Aryl amines are frequently encountered in pharmaceutical research, so harnessing this functional group would also..."

Last sentence in introduction "Collectively, these analyses highlight quantitatively the value that an amine-acid esterification would provide as an addition to the synthetic toolbox."

In results and discussion

2nd paragraph "The substrate scope for this reaction is broad"

We thank the Reviewer for these most helpful suggestions, there edits are captured in the updated version of the text.

Reviewer #3 (Remarks to the Author):

Shen, Mahjour, and Cernak disclose a method to O-arylate carboxylic acids via the use of diazonium salts. A copper catalyst is used, giving the method some similarities to Chan Lam oxidative coupling reactions or copper-mediated Sandmeyer reactions. The developed method is relatively straightforward, occurring in quite high yield with a good scope under mild reaction conditions. A high throughput examination of scope is carried out, with four different diazonium salts being coupled with 96 acids (384 different products), all run in quadruplicate. Select hits are scaled up to provide useful quantities of product and validate the high throughput evaluation method.

The manuscript is well written and fairly easy to read despite the large amount of data disclosed from the HTE data. The science also appears rigorously executed and high quality. There are certainly limitations with the use of diazoniums as coupling partners, but the broad scope investigated suggests that it may on occasion be worth the effort. The incorporation of HTE into the method development and scope evaluation is also seldom done and of general interest. As such, publication in Communications Chemistry is recommended as-is.

We thank the Reviewer for this most supportive feedback, and hope they will find the newly submitted manuscript much improved.

Many thanks for your time and consideration.

Tim Cernak

Assistant Professor
University of Michigan
Department of Medicinal Chemistry, College of Pharmacy
Department of Chemistry, College of Literature, Science and Arts
Program in Chemical Biology, Life Sciences Institute
Michigan Institute for Data Science
930 N University Ave, CHEM 3815, Ann Arbor, MI 48109
+1.734.615.2178, tcernak@umich.edu, www.cernaklab.com

REVIEWERS' COMMENTS:

Reviewer #2 (comments to Authors):

Editorial note: this reviewer provided no further comments for the authors